# Optimization of Edge Resources for Deep Learning Application with Batch and Model Management

**DOI:** 10.3390/s22176717

**Published:** 2022-09-05

**Authors:** Seungwoo Kum, Seungtaek Oh, Jeongcheol Yeom, Jaewon Moon

**Affiliations:** Korea Electronics Technology Institute, Seongnam 13509, Korea

**Keywords:** batch inference, edge computing, edge optimization, deep learning application framework

## Abstract

As deep learning technology paves its way, real-world applications that make use of it become popular these days. Edge computing architecture is one of the service architectures to realize the deep learning based service, which makes use of the resources near the data source or client. In Edge computing architecture it becomes important to manage resource usage, and there is research on optimization of deep learning, such as pruning or binarization, which makes deep learning models more lightweight, along with the research for the efficient distribution of workloads on cloud or edge resources. Those are to reduce the workload on edge resources. In this paper, a usage optimization method with batch and model management is proposed. The proposed method is to increase the utilization of GPU resource by modifying the batch size of the input of an inference application. To this end, the inference pipelines are identified to see how the different kinds of resources are used, and then the effect of batch inference on GPU is measured. The proposed method consists of a few modules, including a tool for batch size management which is able to change a batch size with respect to the available resources, and another one for model management which supports on-the-fly update of a model. The proposed methods are implemented on a real-time video analysis application and deployed in the Kubernetes cluster as a Docker container. The result shows that the proposed method can optimize the usage of edge resources for real-time video analysis deep learning applications.

## 1. Introduction

As the deep learning service spreads over wide areas including video/image analysis, text analysis, or natural language processing, the edge computing architecture gets more focus these days. In edge computing, a trained model and/or application is located on an edge resource, rather than cloud. Though the local (edge) resources have rather limited or restricted computational power when compared to the cloud, edge computing provides many benefits for deep learning applications, especially from the perspective of the cost, such as resources and delay. An edge resource refers to a computation resource that is located close to the data or the service endpoint, and processing on an edge resource can provide faster response, and it can reduce the cost of using cloud resources. However, it is also true that the resources on the edge are restricted. Usually, it is not scalable or extensible like those on the cloud, and is mostly not as powerful. Devices such as Nvidia Xavier with Jetson platform [1] or Google’s Coral TPU [2] are good examples of embedded hardware with acceleration capability, suitable to be used as an edge resource. Due to these differences in computational resources, it is now quite a common practice to differentiate the role of cloud and edge resources in the realization of a deep learning service—powerful and scalable cloud resources for the training of a model, and restricted edge resources for the inference. Many cloud platforms, such as the ones from Google Cloud Platform (GCP) or MS Azure, adopt this concept to provide a deep learning service with cloud and edge. Figure 1 depicts this concept of using cloud-edge resources in realization of a deep-learning service.

The edge computing architecture is now considered one of the major pillars of Industry 4.0, and is widely adopted in various applications from cloud-based IoT to the AI service [3]. Various approaches to realize cloud computing for Industry 4.0 are presented [4]. Object detection is one of the famous use cases in realization of Industry 4.0 with edge computing including defect detection, vehicle detection, and identification. Many neural network architectures and platforms are there for object detection [5], and are improved for use cases, including industrial surface defect detection, laser chip defect detection [6], and fabric defect detection [7].

However, placing AI on the edge resources is still challenging due to the nature of edge resources. It needs to resolve resource heterogeneity, transmission bandwidth with other resources, and most of all, deal with limited resources. Unlike the cloud resources, which are homogeneous and considered infinite, edge resources are limited with its own hardware capacity that is hard to scale, and the platform may differ from each other. Such a stochastic nature of edge may potentially degrade service quality.

Recent edge computing technology adopts solutions from cloud computing and deep learning to deal with those issues through resource scaling, workload distribution, and lightweight models. Larsson et al. have proposed an architecture for applying Kubernetes on multi-edge cluster [8] to handle error conditions and scaling of resources. From the cloud technology, it adopts scaling and scheduling of workload on distributed edge resources. AirEdge [9] proposes a resource orchestration for the edge resources that consists an aerial computing platform. Toka et al. [10] proposed scaling management of Kubernetes edge clusters to maintain quality of service on edge cluster. KaiS [11] proposes a scheduling method for Kubernetes-based edge–cloud cluster. These works approach efficient scaling of edge resources, and Hadidi et al. [12] presented performance evaluation of edge resources on various models from the viewpoint of time, energy, and temperature. Additionally, there is another approach to resolve them by distributing workloads over cloud and edge resources. EdgeLens [13] proposed a framework to distribute workloads on cloud and fog nodes for object detection service, and Vater et al. proposed a modular framework for workload distribution [14]. Further, a single model is split into sub-models and distributed over multiple resources [15]. From the deep leaning side, a lightweight model is applied for edge computing. By applying various methods, such as quantization or pruning, a model can be transformed to the one with a small size and fewer computations [12]. Combined with cloud technologies, those studies provide methods to enable efficient distribution of small-sized workloads on edge resources.

One of the drawback of workload distribution is that it may degrade the quality of service since it needs transmission of data between them. This becomes significant when it comes to a data intensive application, such as video processing or a large amount time-series data analysis. From the viewpoint of efficiency, it would be better if a single workload is located on a single node utilizing as large a resource as it can get. Suppose that there is an edge resource capable of video analysis processing with 10 fps. To increase the performance, it can try to optimize the model itself or distribute workloads on multiple resources to reduce workload for better performance. This paper, on the other hand, proposes a method to increase the workload with batch inference to maximize the usage of existing resources. To this end, the batch inference has been applied to the edge resources to make the optimal use of resources, and then a pipeline architecture for the inference is designed and implemented to verify the effect of batch processing. To see how those methods can be used in the actual cloud–edge environment, the proposed method is implemented as a Docker Container [16] and applied to a Kubernetes [10] cluster which has heterogeneous nodes from cloud and edge. The results show that the proposed methods can provide a real-time analysis service with batching.

The contributions of this paper are as follows:Batch Inference and Its Management: an algorithm to manage the size of input batch for an inference. Unlike other cloud technologies that focus on reducing workload by distribution or scaling, this algorithm is designed to intensify the workload to the maximum, so to make it able to host one application on a single node to reduce communication costs between distributed resources in a cluster.Optimization of acceleration hardware usage: acceleration hardware is not involved in the whole inference cycle, and sometimes it may go idle. The authors identify the process where the acceleration hardware is mostly utilized, and proposed a novel pipeline structure that is able to optimize the usage of acceleration hardware in the batch inference process.Real-time service constraint: the authors analysed the delays in the inference process to determine optimal batch size with respect to both resource availability and real-timeconstraint.The authors demonstrate the proposed method’s performance with the usage of GPU, memory, and the inference delay on an actual Kubernetes cluster in a testbed. The tools for deployment, monitoring, and model management of a deep learning container is implemented, along with a deep learning inference container itself.

The remainder of this paper consists as follows. The background and motives are presented in Section 2, followed by the related works in Section 3. The details of the design are presented in Section 4. The experiment results are given in Section 5.

## 2. Background

As stated in the previous section, this paper focuses on how to manage the size of batch on inference so as to use resources in an optimized way and how to manage them on the cloud–edge cluster. Before presenting the proposed methods, a brief description of the backgrounds is presented in this section.

### 2.1. Embedded Devices with Acceleration

There are many embedded devices that can be used as an edge resource. The characteristics of embedded edge resources are presented in this subsection.

Architecture—aarch, aarch64The embedded systems are not based on the same architecture as resources on the cloud, but using SoC, such as ARM. Some of them have acceleration hardware module integrated in SoC.Shared memoryUnlike a GPU module used on a PC or workstation, that is an independent device and has dedicated memory, on embedded systems a GPU module is integrated as an IP module in an SoC or SoM. The memory is shared for both CPU and GPU modules. Suppose that a device has 16 GB of memory and the CPU consumes most of it like 15 GB. When GPU tries to assign 2 GB of memory to load a model, this will fail with an ‘out of memory’ error. So, for embedded systems with acceleration hardware, care needs to be taken to monitor usage of memory on CPU and GPU.Lightweight deep learning (inference) platformUsually, vendors provide lightweight deep learning platforms for their embedded devices, for example, TensorFlow Lite from Google or TensorRT from Nvidia. The trained models are transformed to corresponding formats to be used on those platforms, and a few optimization methods, such as quantization can be applied in this transformation step. Mostly these kinds of lightweight deep learning platforms provide inferences only. Actually, TensorFlow Lite supports training but it needs to re-design the model to train with TensorFlow Lite.

### 2.2. Pipelining Inference Process

Inference consists of a few processes: pre-processing, inference, and post-processing. In many cases, the input layer of a deep learning model is not the same as the input data, and it needs pre-processing of input data to make it fit the input layer of a model—for example, images need to be resized and transformed to a matrix with fixed dimension to be fed to a model, such as classification or object detection. The same applies to the output layer. The output layer emits information that needs to be interpreted—for example, the data emitted on the output layer of an object detection model, such as Yolo [17], contains locations, classes, and confidence score of detected objects. Jeong et al. [18] have proposed pipelining of these processes to optimize the performance of Jetson devices. Though it is hard to separate those processes as per-clock operations, this pipelining concept is helpful to identify where major delay causes and to optimize the performance with multi threading or parallel processing.

Different resources are used on each process according to its operation. For the inference, the parallel processing capability of a GPU can be utilized, especially when the model is CNN-based. For the pre-processing and post-processing, CPU is used, unless the binaries are built to use GPU for specific operations, such as image resizing.

Batch inference can be a good option for optimization on inference process. Instead of feeding one image to the inference and repeating the next one sequentially, feeding multiple images at once will make the inference delay shorter. For applying the batch inference onto a real-time service needs one more consideration. In the batch processing, the pre-processed inputs are to be held until the batch is ready, which will induce a starting delay. For example, consider an object detection service from a live video stream with a batch size of 4. It needs to wait for the first four frames to come when it is started, so with 30 fps it will have around 12 ms of starting delay.

### 2.3. Cloud–Edge Computing Platform

Edge computing in general is not an antonym or opposite to cloud computing. The purpose of edge computing is to ease the workloads of cloud by offloading them to edge resources, so, in edge computing, both cloud and edge work together to provide a service. In cloud computing, a workload is deployed to a resource from a resource pool that consists of federated resources. Edge computing is almost the same except that the resource pool consists of heterogeneous resources. This heterogeneity brings a few more considerations on edge computing—it needs multiple workload implementation according to the type of the resource, it needs to decide where (to cloud or edge) to deploy the workload, and it needs different metrics to monitor various kinds of resources.

The methods that are proposed in this paper leverage market-leading cloud technologies. To federate resources in a cluster and manage the resources, Kubernetes [19] is utilized. The Docker container [16] is used to make the application deployable with Kubernetes. Additionally, Prometheus [20] is used to monitor the resources on both cloud and edge resources. Details are presented in Section 4.

### 2.4. Resource Monitoring

After resources (either cloud or edge) are assigned for each workload, it is important to monitor those resources to see everything works fine. There are two kinds of metrics to monitor from different perspectives. From the perspective of resource usage, the metrics can be usage of computing resources such as CPU usage, memory usage, GPU usage, or storage usage. From another perspective of application, it also needs to ensure that all the processes meet quality of service requirements. For example, it is important to process the image in less than 3 ms for a service to process 30 fps real-time image.

The resource usage metric can be optimized with the monitoring results. An application container which requires 30% of GPU usage can be deployed to a resource that has 70% of GPU availability. Or if GPU usage goes higher for example 100% for more than a specified period of time, a new resource can be assigned for the deployment of the existing container application. For the second kind of metric, it can be used to monitor and ensure the quality of service. If the requirements are met (takes more than 3 ms to process a frame in the above example), it can re-configure the pipeline to increase fps, for example increasing the batch size or increasing the number of threads for pre-processing.

## 3. Related Works

There is research on the optimization of a deep learning model for edge resources based on microservice architecture. Microservice is a software architecture that builds a service with the combination of loosely-coupled microservices [21,22], and edge resources are utilized to offload workloads of cloud. Due to the rapid improvement of edge resources [23], many video-processing applications, such as real-time object detection [24] and surveillance [25], are using edge these days.

Utilization of edge analysis has been popular in IoT technology area. Edge computing has been applied to process collected data on the edge resources [26]. Then, from the cloud technology, offloading of AI onto edge resources has been investigated. To make a model more lightweight, pruning or quantization has been investigated. Song et al. [27] has proposed pruning to reduce the number of computation and size of a model on deep learning. Quantization is an effective method to reduce the size of a model. TensorRT [28] supports quantization of weights from 32-bit to 16-bit, and even to 8-bit. Courbariaux et al. [29] proposed even further. It builds a model with weights that have only +1, 0, and −1. Those studies focus on reducing required resource footprint to run the model so that it can run on restricted resources. The proposed method can work together with those research. With the lightweight model, there can be more available resources that can be utilized with the proposed methods.

Workload offloading is another topic on realization of edge computing. Edge AI [30] proposed a method that partitions Deep Neural Network (DNN) models and distributes them on the cloud and edge resources. Ko et al. have applied similar model partitioning on DNN model and distribute them on cloud and edge resources. Jiang et al. also distributed workloads on cloud and edge resources [31]. JALAD [32] proposed decoupling of the deep neural network. Goel et al. [33] proposed hierarchical neural network structure that is suitable for edge resource collaboration. Couper [34] used container technology to locate sliced model on edge resources with Kubernetes. Those research focuses on distribution of workloads from cloud to edge resources, with optimization of costs regarding bandwidth and delay for transfer data between them. The proposed work is different from them since it focuses on delivering deep learning model as is to an edge resource.

From the viewpoint of pipeline, there is research to provide frameworks to utilize edge resources. Coello et al. [35] proposed a framework to train a model from image set and provide model serving with custom API. ALOHA [36] proposed a tool set to train deep learning model and application design on embedded heterogeneous architectures. KubeFlow [37] utilizes Kubernetes for building pipeline for training and deployment of trained model in a container. Jeong et al. [18] defined processes that consist of an inference, and proposed optimization methods on a Jetson platform. When a model is trained, it needs to be served. BentoML [38] provides methods to containerize AI model and API to access the model in it. DLHub [39] is proposed as a model hub to store and retrieve machine learning models. InferLine [40] proposed a system to meet latency with minimal cost by using a planner and tuner. PERSEUS [41] is proposed to reduce cost of serving a model. INFaaS [42,43] proposes a model-less serving with automated model variants selection on heterogeneous resources. Gillis [44] proposes methods to serve a model with partitioning. The proposed method provides serving of deep learning inference, but not as a model serving, but as an inference. The proposed methods encapsulate whole inference process to minimize latency due to transmission of intermediate data.

Regarding resource monitoring and scheduling, DART [45] proposed scheduling architecture with data parallelism on heterogeneous resources. Olympian [46] proposed GPU usage scheduling for deep learning model serving. Mabrook et al. analyzed the cost of data analysis on edge resources with a container [47]. Additionally, there was a study that assessed the effect of resources on deep learning model operation [48]. Those studies focused on the scheduling of resources on a lower level, such as hardware resource management. Valentino et al. [49] proposed methods to reduce a cost in a practical way. This paper proposes resource usage on a rather high-level between CPU and GPU, along with batch inference.

## 4. Methods

The proposed methods are described in this section. The proposed method consists of a tool to manage batch size of a running application, and other tools to manage model and monitor resources. In this section, first how the batch inference configured in an inference pipeline is presented, followed by the analysis of delays in the batch pipeline to provide a real-time service. Then, what is needed to deploy the inference pipeline in an actual edge resources are presented, including containerization method, model management, and resource monitoring.

Before going into details of them, the use cases and definition of edge that is used in them for clarity here.

Use case #1: non real-time object detectionThe first use case is an object detection service from a video stream. It selects a few frames from a video stream and build a batch of predefined batch size. The batch input are fed to the inference and the results will be parsed in the post-processing. This use case is suitable for the scenarios with less strict real-time requirements, such as detection of diseases of a plant or detection of weight of a livestock.Use case #2: real-time object detectionThe second use case is almost the same as the first one, except the real-time requirement. It processes all the frames from video input, and generate the output in real-time. A good example of this use case is the surveillance service that needs to detect something right away.Edge device and its locationHere, an edge refers a device that is an embedded device with acceleration hardware. Nvidia Xavier or Google TPU is a good example of an edge device that is used in this paper. Additionally, the location of edge is not on the cloud, but on-site. These are just to assume a kind of extreme edge configuration.

The authors have applied the proposed method on the second use case, to validate the methods are useful to provide a deep-learning based real-time inference service. For a real-time video analysis service, lagging is quite a common problem on resource-constrained edge resources. If the previous frame processing is not completed until the next frame arrives, the output will be lagged and not able to provide a real-time service. Figure 2 depicts this problem: if the inference processing delay becomes bigger than the time between two consecutive frames, the output is to be lagged. Here in this paper, authors have identified a major cause of the lagging, which is the acceleration hardware is not fully utilized. If the inference pipeline can be optimized to increase acceleration hardware usage, the inference delay can be reduced accordingly.

This Chapter consists as follows. Section 4.1 explains how the authors identified resource-intensive processes that consist of an inference and a pipeline architecture to optimize resource usage with batch processing. To avoid the lagging problem, the batch size needs to be decided with respect to the time to build a batch input, and Section 4.2 presents the analysis of those two variables and identifies constraints on batch size for real-time video analysis service. The remainder of this Chapter presents architecture to manage batch size on an edge cluster. In Section 4.3, the author’s previous work on model containerization is introduced briefly, followed by model management for on-the-fly model update, and resource monitoring in Section 4.4 and Section 4.5, respectively.

### 4.1. Pipelining Inference Process with Batch Processing

As described in Section 2, the inference consists of three processes: pre-processing, inference, and post-processing. A very simple implementation to process a video would be adding a for or while loop on top of the sequence. The process is depicted in Figure 3a. Here, the whole processes are iterated in sync with frame inputs. So when the second frame comes, it has to wait until the completion of the first frame, which will generate additional delay. Moreover, the resources are not used efficiently. Note the gap between the end of first frame inference and the start of second frame in the figure. During that period only CPU are engaged to the computation, and GPU will be idle.

In this paper, the authors propose to turn this sequence into a pipeline that consists of processes using GPU and CPU. In the proposed pipeline, the inference process uses GPU while the other process uses CPU. However, unlike the pipeline in the computer architecture, those processes are not easy to be isolated since it is a more high level operation than a CPU operation. So, in the proposed method, a queue is added between each process to separate them from each other. This structure is depicted in Figure 3b. You can see the input and output queue inserted between pre-processing and inference, and inference and post-processing, respectively.

The proposed inference flow is presented in Figure 3c. For simplicity, only the input queue is depicted in the figure. Here, the pre-processing puts the pre-processed input to the input queue. The inference process fetches the input from the queue and do the inference. If the pre-processing is faster than the inference, there will be always data to pop in the queue, which will make the full use of the GPU usage for inference. In the figure, inference fetches new data as soon as it completes previous computation. This will help optimization of GPU usage—compare the figure with Figure 3a, which has a huge gap where GPU resource is idle. Further, the authors propose to add batch concept to each queue. An input batch is fetched from the queue to the inference module when the input size matches the size of batch.

### 4.2. Considerations on Batch Size, Delay, and Real-Time Constraint

Obviously, the batch inference will reduce the time to process data since it needs only one inference for a batch of input data. However, the pre-processing needs to wait some additional time to acquire inputs to build a batch. In this section, the relationship between the delays induced by each process, including batch inference, to provide a real-time video analysis service is presented.

In sequential processing, the total delay for processing an input is sum of pre-processing delay $\left(d_{pre}\right)$, inference delay $\left(d_{infer}\right)$, and post-processing delay $\left(d_{post}\right)$.
(1)$$d_{seq}=d_{pre}+d_{infer}+d_{post}$$

Additionally, the processing delay of multiple inputs with sequential processing can be calculated with the following equation, where *N* denotes a number of images.
(2)$$d_{mul\_seq}=N\times d_{seq}$$

Let us take a look on batch processing delay. For a batch processing, images are to be collected until batch size $N_{batch}$, and then pre-processed. In this case, the pre-processing delay will be $d_{pre}*N_{batch}$. The batch inference will generate an additional delay which depends on the size of batch, which is denoted as $d_{batch\_inf}$. The post-processing delay is a little bit different. It can be assumed that post-processing delay would be the same of that of pre-processing, which is N times of single post-processing delay. However, unlike the pre-processing, where all the parameters such as size of the image or array are consistent, the computational load of post-processing varies due to the model structure. For example, in object detection model, such as Yolo, the post-processing processes data from the output layer of a model to find classes, confidence, and location of the detected objects in the image. If there are many detected objects in an input image, this will take longer. If there is no detected objects then this post-processing will do nothing. Thus, the delay of post-processing in batch $d_{batch\_post}$ inference should be treated independent from $d_{post}$. Considering these delays, batch processing delay can be represented as the following equation.
(3)$$d_{batch}=N_{batch}\times d_{pre}+d_{batch\_infer}+d_{batch\_post}$$

The idea is that $d_{batch}$ is to be equal or less than the $d_{mul_{s}eq}$ since it uses parallel processing capabilities of hardware acceleration. The test results verify this idea, which is given in the next section.

There is one more consideration if the service is to process data in real-time. It needs to be able to process input data before the next input data arrives to support a real-time processing of continuous data input. Suppose that we have a video input with 30 fps. To avoid frame drop or lagging, each frame needs to be processed in 1/30 s. Additionally, the condition for real-time processing with respect to delay can be presented as follows.
(4)$$t_{interval}<d_{seq},\;\mathrm{for}\;\mathrm{single}\;\mathrm{data}\;\mathrm{processing},$$
(5)$$t_{interval}\times N_{batch}<d_{batch},\;\mathrm{for}\;\mathrm{batch}\;\mathrm{processing}.$$

### 4.3. Containerization of a Model and Its Application

To make it able to deliver a batca inference application, the Docker container is leveraged in the proposed method. As presented in Section 3, Container is widely used for deployment of various Machine Learning applications. Here, the proposed method is implemented as a container, with APIs to manage the batch size for the optimization of resource usage. The authors have proposed a containerized AI microservice [50] that exposes APIs to access the AI methods in it. Here, that container method is extended to provide one additional interface to manage batch size.

### 4.4. Management of Models with Different Batch Size and Live Model Update

Batching is a kind of standard procedure for the training a model, but it is not quite popular in inference or prediction. Actually, there are a few literatures describing using batch for the inference, but many of them do not refer to methods that utilize parallel processing of GPU devices. Some of them utilize multiple inference threads for the batching, which can affected by the operating system. Some of them refer to loops or even use only the CPU. To eliminate the effects of resource usage other than GPU’s parallel processing for batch inference, in this paper the authors built each model that has a different size of input layer with respect to the batch size. For example, if a model has 128 × 1 size of input, a model with batch_size = 2 will have 2 × 128 × 1 input layer.

The proposed method is to manage the batch size according to the resource availability, and it needs to update the model to a new one with new batch size when it is changed. The proposed method includes a tool for this model update. The AI Model Repository (AMR) stores a trained model with specific metadata, including model name, model version, and size of batch, and supports query and retrieval of the model. This can help reduce service downtime due to model substitution. The containerization API to change a batch size mentioned in Section 4.3 will communicate with AMR to retrieve a model with a new batch size.

### 4.5. Edge Resource Monitoring and Optimization

This subsection describes the monitoring of GPU resources, and optimization of resource usage combining the components described in the previous sections. This paper describes optimization of GPU resources. For the monitoring of GPU resource usage, it depends on the libraries and APIs that are provided by the device vendor. Nvidia provides APIs to find the usage of their own GPU devices such as nvidia_smi or tegra_stats for Tegra-based devices. When the usage of a GPU is measured, it needs to be collected for monitoring. Prometheus is a monitoring tool for resource usage and very flexible to adopt various metrics. The proposed method in this paper leverages those technologies to monitor resource and configure batch size accordingly.

Figure 4 depicts the architecture of the proposed management that consists of the four major modules deployed in the following resources.

Edge Resource: A Resource Usage Collector (RUC) is running on an Edge resource, to collect the resource usage, mostly of GPU, and transfer the collected usage information to Resource Usage Monitor (RUM). A RUC provides a few configuration variables, including target resource, units, and their collecting frequencies. For example, an RUC can be configured to monitor GPU resource usage in percent for every five seconds.Edge Usage Optimization Service: Resource Usage Monitor (RUM) retrieves collected resource usage data from each edge resource. Basically, this RUM is a database. The Adaptive Batch Controller (ABC) retrieves the collected data from RUM and decides the new batch size for each AI application running on an Edge resource. The new batch size is then transmitted to an AI application to invoke loading of a new model with a new batch size.AI Model Repository: This AI Model Repository (AMR) is a file-based database of AI model files. It stores models that have the same network with different batch size. It provides a query API with model name, version, and batch size to make it able to find and download a model of interest.

One of the important parts of the proposed method is how to decide a batch size. The batch size needs to be selected with caution to prevent bottlenecks and reduce service downtime. In the proposed method it takes three considerations on batch size decision—available memory for GPU, GPU usage, and real-time criteria. The last one is more related to the application level metric, rather than the resource metric. Suppose that only resource metrics are used in batch size decision. In this case, it will try to reduce the batch size when GPU consumption hits 100%. However, in the proposed method it keeps batch size as is as long as the real-time criteria is met (Equation (Equation 5)). The algorithm is presented in Algorithm 1. The algorithm takes considerations from perspectives of both resource and application metrics. First, it sees whether the real-time criteria is met (line 5) and try to increase the batch size. If it has available resources, it grants the new batch size (line 9). If the memory is not sufficient to load a new batch size model, it will stay the same batch size (line 12), or the current batch size cannot guarantee the real-time criteria, it will decrease the batch size (line 15).
**Algorithm 1:** Batch Size Decision1:**Data:** BatchSizeArray[N] // An array that stores batch size2:**for** each processing **do**3:      Collect resource usage and store it to mem_used4:      Collect processing delay and store it to delay5:      **if** delay < real time criteria and *n* < *N*
**then**6:          temp_batch_size = BatchSizeArray[n+1]7:          **break**8:          **if**
mem_used < mem_threshold_ratio
**then**9:              new_batch_size = temp_batch_size10:            **break**11:         **else**12:             continue13:         **end if**14:     **else**15:          new_batch_size = BatchSizeArray[n−1] unless new_batch_size is non-negative16:     **end if**17:**end for**18:**if** 
new_batch_size 
**then**19:     Break and load new model with new batch size20:**end if**

## 5. Experiments and Results

The proposed work is implemented on a testbed for validation. The implementation was completed on actual devices and applications to provide more ready-to-adopt technology, rather than proposing laboratory level proof-of-concept. The actual cluster that consists of both cloud and bare-metal edge resources is implemented as follows. First, the resources—a virtual machine (Cloud RSC) in Azure cloud platform and three edge resources are prepared. The first edge resource (Edge RSC #1) is a baremetal device with Intel i7 CPU, 32 GB memory and Nvidia 2080ti GPU (16 GB dedicated memory), and the other two resources (Edge RSC #2 and #3) are Nvidia AGX Xavier developer kits, which has Tegra-based ARM module with 32GB shared memory and 32 GB internal storage. The storage is extended to 1 TB with NVME m.2 SSD. All the resources are running on Ubuntu 18.04. To provide cloud-native environment, a Kubernetes cluster that consists of those resources is constructed. Kubernetes version 1.22.4 are employed in the construction of the cluster.

Software is prepared according to the architecture depicted in Figure 4. Two containers—one for AI Application and the other one for Resource Usage Collector—are built. The AI Application provides object detection deep learning service with Yolo v4 models. The Yolo v4 application is containerized in a container, which provides RESTful interfaces for the management of batch_size. The interface definitions follow the previous work [50].

For the RUC and RUM, the Prometheus is used. RUC is a container that is an instance of Node Exporter of Prometheus, while RUM is an instance of Prometheus Server. The metrics of RUC are configured for the GPU and Memory usage. Note that, for Edge RSC #2 and #3, which are embedded ARM SoC devices, the memory is shared between GPU and CPU. RUC is built as two containers with respect to the architecture (AMD container for Edge RSC #1 and ARM container for the others) and deployed on the Kubernetes cluster with DaemonSet resource. The RUM is installed on a Cloud RSC. Prometheus exporters are configured to collect GPU resource usage and memory used by GPU every second.

AMR is implemented as a file database, based on GridFS. It provides an API to store a model with the model name, model version, and size of batch, and another API that queries a model with the same arguments and downloads it. AMR is installed on a VM on MS Azure cloud instance.

The algorithm is implemented in ABC. It reads the collected metrics from PromQL database of Prometheus in RUM, decides a new batch size and call the API of AI Application deployed on the cluster.

To manage of deployment of a container to a node in a cluster, a Container Deployment Service (CDS) is implemented. It provides a GUI that displays the nodes in a cluster and container images in a container repository, and supports drag-and-drop user interfaces for the deployment of a container to a specific node, along with a button to call the APIs of the deployed container.

Figure 5 depicts the configuration of the actual testbed, and Figure 6 depicts implementation of CDS.

The AI container works as follows. It loads the default model (with batch size = 1) when it is initiated, and loads a sample video file then applies the input to the Yolo v4 model. The metric (GPU and memory usage) is collected and transmitted with RUC. ABC decides the new batch size based on the proposed algorithm and transmits the value to the running containers. When a new batch size is received from ABC, it loads a model with new batch size from AMR, and then applies the same sample video file.

The model files are prepared with batch sizes 1, 2, 4, 8, and 16. Each model is built from the same Yolo v4 model with extended input layer to match the batch size and transformed to a TensorRT model.

Table 1 shows the delay measured on AGX Xavier Edge Resources with batch size configurations 1, 2, 4, 8, and 16. The video input used was a video file with 1080 × 720 resolution, encoded with H.264 and AAC codecs in 30 fps. The delay for each step (pre-processing, inference, and post-processing) are tracked for different batch sizes from 1 to 16. The delay of processing batch inputs in each process is given on the first, second, and third rows. Total delay for a single batch processing is given on the fourth row. As given in Equation (Equation 3) in Section 4.2, it can be seen that the pre-processing delays are increased almost linearly proportional to the batch size, while the inference delay is not linearly proportional due to parallel processing of GPU or hardware accelerator. For the post-processing which the number of detected objects has more effect than the number of images, does not relative to the batch size. On the fifth and last row of the same table, the real-time constraints can be found. The fifth column shows real-time conditions to build and process a batch. For example, to build a batch with size 4, it needs 0.1333 s. In other words, to process a batch with size 4 in real-time, it needs to be able to process it within 0.1333 s. The table shows how much margin can be provided for real-time processing with different batch sizes. The results of the same test on Edge RSC #1 are given on Table 2. Since it has more powerful acceleration device, the table shows that more margin for real-time processing is guaranteed with batch processing.

As the effect of the batch inference has been identified with the experiment, the algorithm has been applied to the testbed. It reads the GPU usage within last one minute from PromQL of the RUM, and then decide the new batch size with respect to the available resources. However, even the proposed work utilized on-the-fly model change, which has much less delay than re-deploying a new container, it was found that the interruption of a service due to this model change is significant.

The usage of GPU resource and memory resources are depicted in Figure 7 and Figure 8, respectively. Figure 7 shows different GPU usage pattern according to the batch size. With smaller batch size, the individual GPU consumption ratio is rather low, and GPU consumption occurs very frequently, as can be seen on the left side of the figure. As the batch size grows, the individual GPU consumption ratio goes high. It can be found that almost all the batch inference consumes more than 90% of the resource. However, the consumption occurs less frequently. With the batch size = 16, it becomes very sparse. Note that the gaps between each batch size change. This gap is the service downtime that is induced by a model change. In our experiment, it took less than 25 s to download a model from AMR and load it onto memory. The size of model is 174 MB. This delay is way less than the delay from re-deploying a new container. In our testbed it took more than one minute to stop and reload a GPU container, even the container is pre-downloaded.

The memory consumption is depicted in Figure 8. It can be seen that the memory used by GPU is increased as the batch size increases.

## 6. Conclusions

In this paper, the authors propose a method to optimize usage of edge resources on running a deep learning application with batch inference. The goal of the proposed method is to optimize usage of edge resources by changing batch size with respect to the available resources. If the resource is available, increase the size of batch, if not then decrease the size of batch. To enable batch size configuration, the processes of which consists an AI inference application and how the resources are used are identified, followed by analysis of delay on an inference application.

The proposed method consists of a few components to realize optimization of edge resource usage. First it needs to know how much resources are available. RUM is responsible for this collecting of resource usage. The decision to change batch size is made in ABC, and a request to change batch size is transmitted to the AI application container on an Edge resource. To enable handling of batch size change, an AI container structure with RESTful APIs are proposed along with AMR which stores and provide models with different batch sizes. Upon receiving the request, the AI application stops running current task, download a model with a new batch size from AMR, and resume the task.

The proposed method is implemented on a testbed, with actual cloud and edge resources. Edge resources includes an embedded device, Nvidia AGX Xavier, to verify that the proposed method can be applied to more practical use case in smart factory or smart farm. A real-time object detection is chosen for the use case to see the effect of model change. For the test, TensorRT platform is used for the inference engine, with Yolo v4 model. Each model with a batch size is built to have different size of input to fit the size of batch and then transformed to TensorRT models. This is to reduce effects from processes other than batch inference.

The result shows that the usage of an edge resource can be optimized with the proposed method. By changing the size of the batch with respect to the available resources, it can increase the usage of it to the optimum level. The proposed method enables real-time processing on an edge resources, on the condition that it has sufficient resources to process a batch input. Two perspectives from resource and application level metrics are used on batch size decision, which makes it able to use the resources to the limit as long as it meets the real-time criteria, which is the time earned by building a batch. Further, the results show that the service downtime due to model substitution can be reduced with on-the-fly update of a model, with the help of AMR. The proposed method can enhance utilization of Edge resource on various services when it is combined to the model optimization, such as pruning or binarization.

In the future, the authors plan to apply enhance the adaptive batch size decision by applying deep learning algorithms. The empirical resource usage data collected on the proposed RUM can be considered as a time-series data of resource utilization, which is a valuable asset to analyze and predict workload of various AI or deep-leaning based inference applications. The batch size decision can be improved for the accuracy when more data are collected and applied to a deep learning algorithm for time-series analysis. Further, more applications other than object detection will be applied and distribution of workload on multiple edges are to be investigated.

## Figures and Tables

**Figure 1 sensors-22-06717-f001:**
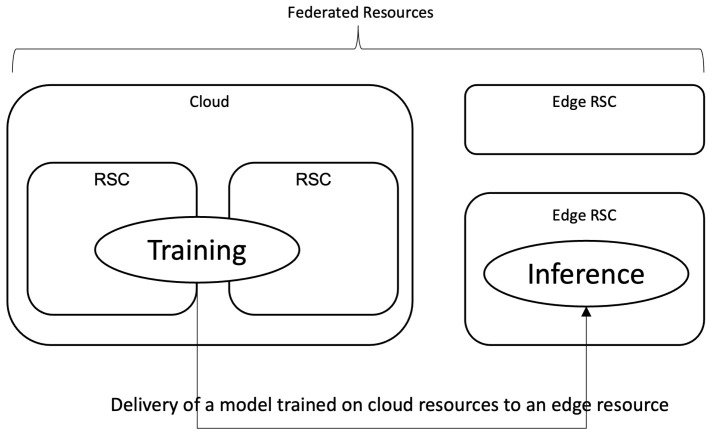
Cloud and Edge resources for a deep learning service.

**Figure 2 sensors-22-06717-f002:**
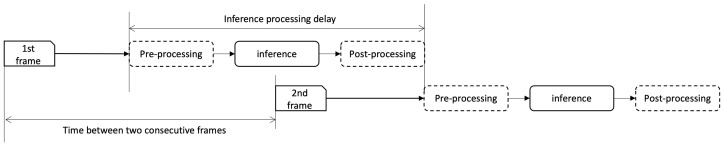
Real-time video analysis application. If inference processing delay is bigger than the time between two consecutive frames, the output will be lagged.

**Figure 3 sensors-22-06717-f003:**
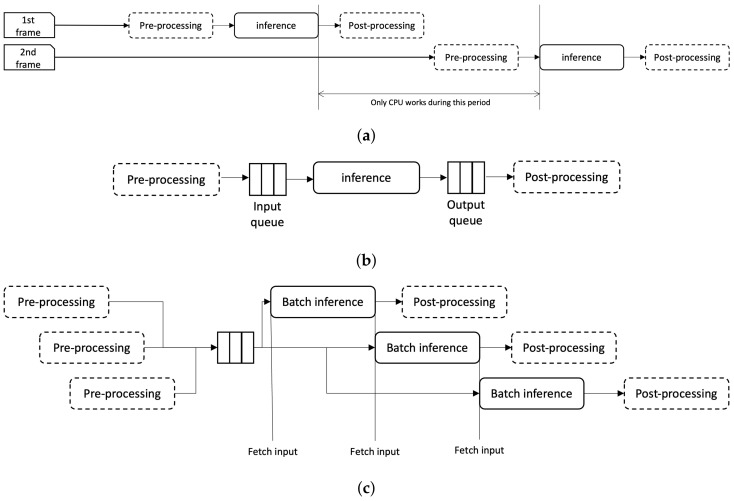
Inference pipelines. (**a**) Sequential processing of a video input. (**b**) Pipeline of sequential processing. Each process is separated with a queue. (**c**) Batch inference pipeline. Input data are stacked on a queue, and inference fetches a batch of images. Lined boxes refer processes that make use of GPU while dotted boxes refer those of CPU.

**Figure 4 sensors-22-06717-f004:**
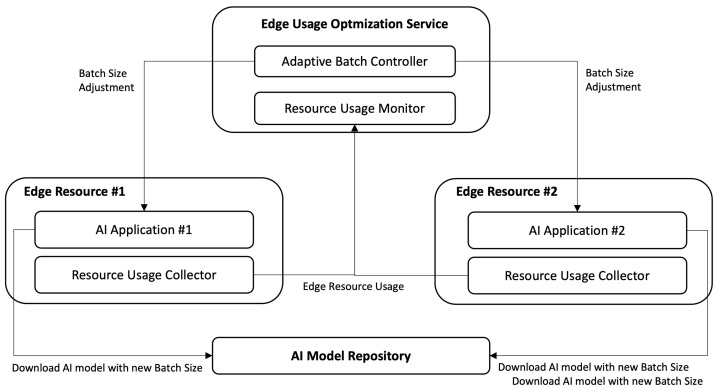
Structure of the proposed architecture.

**Figure 5 sensors-22-06717-f005:**
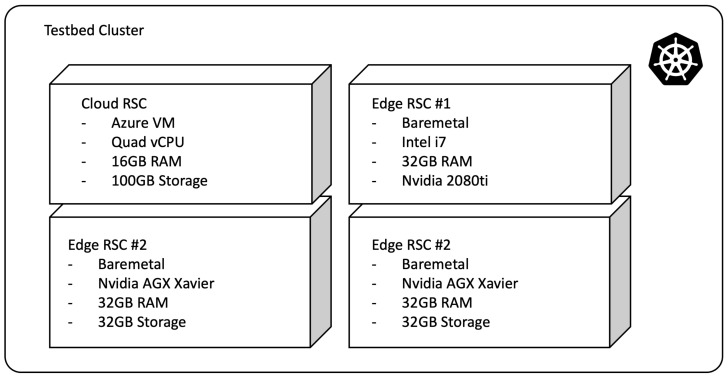
Testbed Configuration.

**Figure 6 sensors-22-06717-f006:**
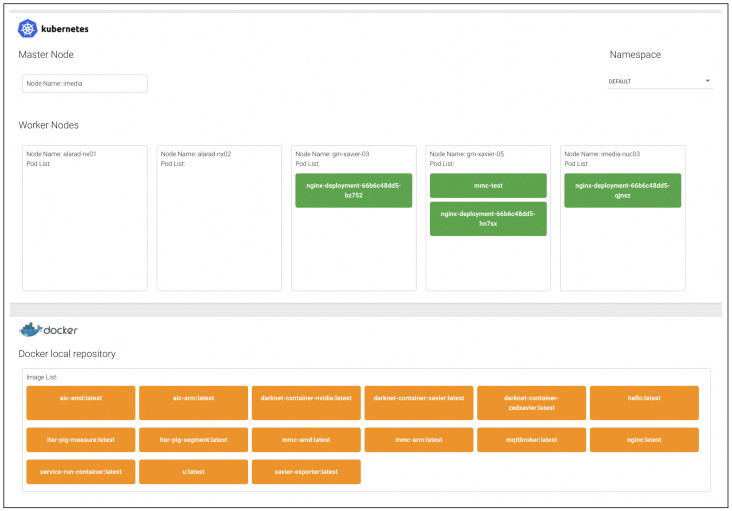
GUI of CDS.

**Figure 7 sensors-22-06717-f007:**
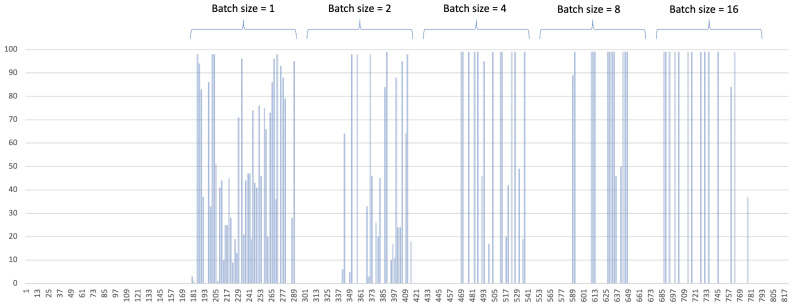
GPU resource usage with different batch size.

**Figure 8 sensors-22-06717-f008:**
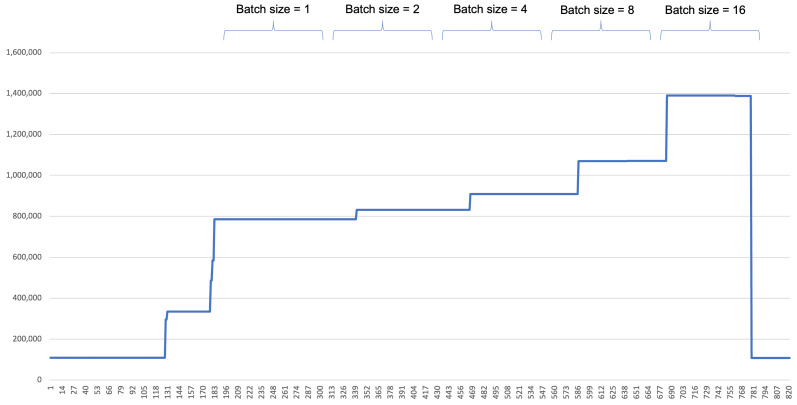
Memory consumption of GPU with different batch size.

**Table 1 sensors-22-06717-t001:** Delays and real-time constraints on a Xavier.

Batch Size	1	2	4	8	16
Pre-processing	0.004	0.007	0.013	0.026	0.051
Inference	0.021	0.031	0.058	0.107	0.211
Post-processing	0.003	0.004	0.008	0.014	0.027
**Total batch delay**	**0.031**	**0.048**	**0.089**	**0.166**	**0.324**
Real-time condition	0.033	0.067	0.133	0.267	0.533
**Real-time margin**	**0.002**	**0.019**	**0.044**	**0.101**	**0.209**

**Table 2 sensors-22-06717-t002:** Delays and real-time constraints on a 2080 ti.

Batch Size	1	2	4	8	16
Pre-processing	0.003	0.006	0.013	0.027	0.053
Inference	0.003	0.005	0.01	0.017	0.035
Post-processing	0.001	0.002	0.003	0.006	0.009
**Total batch delay**	**0.011**	**0.021**	**0.042**	**0.82**	**0.163**
Real-time condition	0.033	0.067	0.133	0.267	0.533
**Real-time margin**	**0.011**	**0.046**	**0.091**	**0.185**	**0.370**

## Data Availability

Not applicable.

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
