# Peer review of "Optimization of Edge Resources for Deep Learning Application with Batch and Model Management"

_sensors, 2022, doi:10.3390/s22176717_

Round 1
Reviewer 1 Report
The manuscript attempts to present a case about optimization of edge resources for deep learning application with batch and model management. The paper is well-written and has the style and language demanded for a potential publication. Some clarifications should be made, along with some corrections here and there, but I believe that the paper will be ready for publication upon the conduction of the aforementioned corrections.
My points are analytically listed below
Points for consideration:
• Point 1: By reading the first paragraph of the introduction, I believe that the author “rushes” too much to get to the main points of the paper. I suggest that a paragraph should be inserted writing down about the Industry 4.0 and its core technologies like the ones written (ML, DL etc.). You should find the following papers helpful and you should include them in the references:
· 10.1155/2022/5023011
· 10.1155/2021/5554685
• Point 2: I strongly suggest to add more references in the list, the total number is too small.
Author Response
First of all, the authors would like to appreciate the author's valuable comments. The authors has been tried to address the comments as follows:
Point 1: By reading the first paragraph of the introduction, I believe that the author “rushes” too much to get to the main points of the paper. I suggest that a paragraph should be inserted writing down about the Industry 4.0 and its core technologies like the ones written (ML, DL etc.). You should find the following papers helpful and you should include them in the references: 10.1155/2022/5023011, 10.1155/2021/5554685
Response 1:
The authors have revised the introduction to follow the comments. Three paragraphs that describe Industry 4.0 and the core technologies are given. Those can be found from line 50. Also, one more paragraph is added to present the existing approach in the same chapter, which can be found on line 64. On those paragraphs, the authors tried to explain the importance of using edge for Industry 4.0, what are the issues on edge computing and how there dealt with, and what is the difference of the proposed work.
The authors concludes the introduction emphasizing the contributions on line 100.
The authors have found that the two papers suggested from the reviewer is really valuable to explain the value of the proposed work, and added them in the reference. Those to references are included in number 3 and 4, which can be found on line 571 and 574, respectively.
Point 2: I strongly suggest to add more references in the list, the total number is too small.
Response 2 : The authors agree that the number of references are not enough to explain the current state of edge computing and how it needs to be addressed, and added more references to the paper from the survey. Most of the newly added references are used in the introduction chapter, to describe the current status of edge computing in recent literatures. Total number of references are increased to 50.
Other modifications: The authors fixed a few typos. All the modifications including the above Responses are tracked and marked with blue color in the revised version.
Reviewer 2 Report
The is a decent piece of work where the authors propose a method to optimise usage of resources on edge devices via adaptive batch size at inference time. They use a Kubernetes instance to demonstrate performance and run their experiments.
I think there are two areas of improvement:
a) Introduction is a bit shallow and blunt; it has to be punchier, further highlighting the importance of optimisation of DL methods for efficiency on edge devices - the section should clearly articulate the contributions with respect to the current methods
b) A use case on object detection is utilised but that is very briefly described - perhaps a figure or two of the problem (albeit not the main focus) could be appropriate. For instance how adapting the batch size affect performance? Object detection is considered across multiple objects or single?
Author Response
First of all, the authors appreciate valuable comments of the reviewer. We have tried to address the comments as the following.
Point 1 : Introduction is a bit shallow and blunt; it has to be punchier, further highlighting the importance of optimisation of DL methods for efficiency on edge devices - the section should clearly articulate the contributions with respect to the current methods
Response 1: The authors agree that the introduction was not giving the status of current approaches and the contributions of the proposed method. The authors revised the introduction to include a few paragraphs describing what are the current state and issues on edge computing. Those can be found from line 48. And then added one more paragraph to describe what is the difference of the proposed work, which can be found from line 84.
Also, the authors has added one more paragraph to clearly present the contributions of the proposed work at the end of the introduction, which can be found from line 101.
Point 2: A use case on object detection is utilised but that is very briefly described - perhaps a figure or two of the problem (albeit not the main focus) could be appropriate. For instance how adapting the batch size affect performance? Object detection is considered across multiple objects or single?
Reponse 2: The authors agree that the previous draft was not able to identify what is the problem and how it can be resolved with the proposed method. A figure (Figure 2) has been added to describe what is the problem that we want to resolve in real-time video analysis, which can be found right before line 287.
Also, how the batch size management can affect the performance is described in a paragraph in line 287. We have tried to explain this idea as a lagging problem. In our paper, object detection is considered across multiple objects, as in Yolo v4 implementation that is referenced in the paper (reference number 17).
Other modifications: The authors fixed a few typos. All the modifications including the above Responses are tracked and marked with blue color in the revised version.